# Tobacco, electronic nicotine delivery system, nicotine replacement therapy, and cannabinoid use during pregnancy: A descriptive cross-sectional survey

Sreemanjari Kandhasamy[1,2], Stéphanie Baggio[1,3], Jérôme Mathis[4], Yolanda Mattmann[4], Emeline Maisonneuve[1,5], Reto Auer[1,6], Alice Panchaud[1,7], Stephen P. Jenkinson[1,6☯], Anna Schoeni[1,8☯]*

1 Institute of Primary Health Care (BIHAM), University of Bern, Bern, Switzerland, 2 Graduate School of Health Sciences, University of Bern, Bern, Switzerland, 3 Institute of Psychology, University of Lausanne, Lausanne, Switzerland, 4 Department of Obstetrics and Gynecology, Centre Hospitalier Bienne, Bienne, Switzerland, 5 Materno-Fetal and Obstetrics Research Unit, Department "Woman-Mother-Child", Lausanne University Hospital, Lausanne, Switzerland, 6 Center of Primary Care and Public Health (UNISANTÉ), University of Lausanne, Lausanne, Switzerland, 7 Service of Pharmacy, Lausanne University Hospital and University of Lausanne, Lausanne, Switzerland, 8 Center for Child Health Analytics (CCHA), Children's Hospital Central Switzerland, Lucerne Cantonal Hospital, Lucerne, Switzerland

☯ These authors contributed equally as the last authors.
* anna.schoeni@unibe.ch

## Abstract

### Introduction

Tobacco smoking is associated with adverse health outcomes for both pregnant women and their offspring. Smoking cessation counseling is an effective method to help women quit smoking. Developing a targeted smoking cessation intervention could benefit those who struggle to quit tobacco and potentially reduce the harm due to any co-occurring tobacco use. Assessing the prevalence of tobacco, electronic nicotine delivery systems (ENDS), nicotine replacement therapy (NRT), and cannabinoid use in pregnancy is key to developing such interventions. Thus, we aimed to assess the prevalence and patterns of tobacco, ENDS, NRT, and cannabinoid use in pregnancy. We further aimed to assess the prevalence of smoking cessation counseling intervention.

### Materials and methods

We conducted a cross-sectional survey among pregnant women attending regular clinical visits at Spitalzentrum Biel between February and May 2023 (n = 262). Frequency and proportion along with 95% confidence intervals (CI) were reported for tobacco, ENDS, NRT, and cannabinoid use in pregnancy.

**Data availability statement:** All relevant data are within the manuscript.

**Funding:** This work has been supported by an Investigator Initiated Clinical Trial (IICT) grant from the Swiss National Science Foundation (SNSF) IICT_33IC30_173552, the Swiss Tobacco Prevention Fund (TPF) 19.017477, and Swiss Cancer Research (SCR), KFS4744-02-2019 (all 3 to Prof. Reto Auer). The funders had no role in the design of the study; in the collection, analyses, or interpretation of data; in the writing of the manuscript, or in the decision to publish the results.

**Competing interests:** The authors have declared that no competing interests exist.

## Results

Tobacco use was reported among 7.6% (20/262, 95% CI: 4.2%−11.1%) of the included pregnant women. Tobacco cigarettes (conventional or roll-on) were used by 7.3% (19/262, 95% CI: 3.8%−10.7%) of the surveyed pregnant women, with 0.8% (2/262, 95% CI: 0.0%−3.4%) of them reporting use of cigarettes along with ENDS and 0.4% (1/262, 95% CI: 0.0%−3.8%) reporting use of the cigarettes with NRT. Cannabinoid use was reported by 3.8% (10/262, 95% CI: 1.1%−7.0%) of pregnant women and all of them used products with Cannabidiol (CBD) only. Additionally, only 25% (5/20, 95% CI: 10.0%−48.3%) of tobacco users had received smoking cessation counseling intervention.

## Conclusion

The estimated prevalence of tobacco, ENDS, NRT, and cannabinoid use among the pregnant women in this survey was 7.6%, 0.8%, 0.4%, and 3.8% respectively. However, among tobacco users, only one-fourth received smoking cessation counseling intervention.

## Introduction

Smoking tobacco in pregnancy is a prevalent, modifiable risk factor for infant mortality and morbidity [1,2]. The global prevalence of smoking during pregnancy was estimated at 1.8%, with Europe reporting a prevalence of 8.1% [3]. In the USA, 5.4% of women who gave birth in 2021 reported smoking while being pregnant [4]. In 2022, 8.3% of women giving birth in Australia admitted to smoking during the first 20 weeks of pregnancy, with 71% of them continuing to smoke throughout their pregnancy [5]. Smoking during pregnancy is associated with an increased risk of several adverse maternal and neonatal outcomes, such as placenta previa, placental abruption, premature rupture of membranes, stillbirth, preterm birth, and small for gestational age [6–10]. However, smoking cessation can help reduce these adverse effects [11–13].

Smoking cessation counseling has been beneficial in supporting pregnant women to quit tobacco use, but some might need additional support with smoking cessation aids to quit smoking [14–16]. In addition to nicotine replacement therapy (NRT) and pharmacotherapy (Varenicline or Bupropion), women view electronic nicotine delivery systems (ENDS) as an aid for smoking cessation during pregnancy [17]. Indeed, a randomized controlled trial (RCT) observed ENDS to be more effective when compared to NRT for smoking cessation, showcasing the potential for promising smoking cessation support in pregnant tobacco users [18]. However, there is limited evidence on the safety of using ENDS and NRT in pregnancy [19–23]. Yet, women who find it difficult to quit tobacco continue to use ENDS or NRT throughout pregnancy. In the UK, pregnant smokers struggling to quit tobacco are recommended to use tobacco-free alternatives such as ENDS and NRT as smoking cessation interventions [16,24]. Notably, ENDS are perceived as a "safer alternative to tobacco" by pregnant

women, which has led to a substantial increase in the use of ENDS during pregnancy [19,25]. The Behavioral Risk Factor Surveillance System Survey, a large health survey that included adult women between 18 and 44 years, was conducted in the USA and observed that ENDS use in pregnancy had almost doubled between 2016 and 2018 [26].

Furthermore, cannabinoids have also been reportedly consumed in pregnancy to treat issues such as nausea, vomiting, increase in body weight, sleep issues, stress, anxiety, depression, post-traumatic stress disorder, and chronic pain [27,28]. The data from the cannabis policy study conducted between 2019 and 2021 on individuals aged 16–65 years in the USA and Canada highlighted that approximately 20% of surveyed pregnant women used CBD-containing products [27]. In 2022, a national survey conducted in the USA on substance use observed that around 7.9% of the pregnant population consumed cannabis (which contained a mixture of THC and CBD) [29]. Cannabinoids like CBD and THC are commonly smoked/vaped and also consumed in the form of oils, edibles, and topical skin preparations [30,31]. When cannabinoids are smoked, they are more likely to be co-used/mixed with tobacco, resulting in an increased risk of adverse neonatal outcomes, thus limiting evidence on the safety of the use of cannabinoids in pregnancy [31–36].

Given these observations, developing personalized, evidence-based, targeted interventions for smoking cessation could potentially provide better support to those struggling to quit/reduce tobacco use and also reduce the harm associated with any co-occurring tobacco use [4,23,26,37–40]. Nonetheless, to develop such targeted interventions, understanding the current prevalence of tobacco, ENDS, NRT, and cannabinoid use among pregnant women is essential. Although nationwide surveys conducted in Switzerland until 2012 indicated that around 10% of women smoked during pregnancy, current evidence on tobacco, ENDS, NRT, and cannabinoid use among pregnant women in Switzerland is limited [41–45]. To bridge this gap, this cross-sectional survey was designed with the primary objective of estimating the prevalence and patterns of tobacco, ENDS, NRT, and cannabinoid use among pregnant women attending regular clinical visits at Spitalzentrum Biel in Switzerland. The secondary objective was to estimate the intention to quit using tobacco among pregnant users. The survey also aimed to assess the prevalence of smoking cessation counseling intervention.

## Materials and methods

### Study design

This was a cross-sectional monocentric survey.

### Setting

The data for the study were collected from pregnant women coming to Spitalzentrum Biel, a public hospital in Switzerland, between February and May 2023. The data used in this study were de-identified and completely anonymized. Thus, the ethics committee of the canton of Bern waived the requirement for a formal submission, as the project did not fall under the scope of the Swiss Federal Human Research Act (HRA), Article 2, Paragraph 1 (Req-2022–01416). Written informed consent was not required due to the collection of anonymized data under HRA. However, mid-wives explained the study to the participants and only collected data from those who had provided verbal consent to participate in the survey.

### Participants

The participants included in the survey were pregnant women who came in for a regular clinical visit (excluding emergency) at Spitalzentrum Biel between February and May 2023. Each participant was eligible to take the survey once. To ensure this, at the beginning of each survey, participants were questioned if they had already taken part in the survey ("*Befragung bereits bei früherer Visite durchgeführt*" translated as "survey already conducted during the previous visit"). Only those who answered "no" to this question were eligible to continue the survey.

## Outcomes

The primary outcomes of the study were the frequency and proportion of pregnant women using tobacco products (tobacco cigarettes (conventional cigarettes or roll-on cigarettes) cigars, pipes), tobacco heating systems (like IQOS, Glo™, Ploom), or smokeless tobacco (like snus) and the use of ENDS, NRT and cannabinoid (cannabidiol only (CBD) or with tetrahydrocannabinol (THC) >1%) products.

The secondary outcomes included frequency and proportion of i. pregnant women intending to reduce/quit/have no change to the current use of tobacco products, ii. pregnant women receiving smoking cessation counseling along with a descriptive summary of suggested alternatives (like NRT, ENDS, pharmacotherapy (Bupropion/Varenicline), and other).

## Demographic characteristics

Information on the maternal age and the gestational age (weeks of pregnancy) was collected from the pregnant women at baseline.

## Instrument

The information for this survey was collected through a questionnaire. The data were collected by midwives who filled in the questionnaires, and the collected data were irreversibly anonymized. The collected information included demographic information (such as maternal age and gestational age), information on the use of tobacco, ENDS, and NRT products in the last 7 days, types and frequency of use of these products per day/ week, intent to quit using these products during pregnancy, receiving smoking cessation counseling, and cannabinoid use in the last 30 days.

## Statistical analysis

Descriptive analysis was conducted to summarize baseline characteristics and to report the primary and secondary outcomes. Means and standard deviations were reported for the maternal age. Median and interquartile range (IQR) were reported for the gestational age of the women. The frequency and proportion, along with 95% confidence intervals (CI), were reported for all the outcomes and were stratified based on the trimester of pregnancy. The 95% CI was calculated using the multinomCI function in R from the DescTools package. All the analyses were conducted with R version 4.3.2 [46].

# Results

## Baseline characteristics

For this survey, 337 completely anonymized responses were collected at Spitalzentrum Biel. Among the collected responses, 262 responses were included in the analysis, while 75 incomplete responses were excluded from the analysis. The excluded responses included a "yes" to the question "Befragung bereits bei früherer Visite durchgeführt" which translates to "survey already conducted during the previous visit". The average age of the included pregnant women was 31.35±5.11 years. More than half (55.3%,145/262) of the included women were in their third trimester (>28 weeks of pregnancy), and the rest were in their second trimester (between 13–27 weeks of pregnancy). The median gestational age of women included in this survey was 34.9 (IQR: 18.5–37.2) weeks.

## Prevalence of use of tobacco, ENDS, NRT, and cannabinoid in pregnancy

In this survey, 7.6% (20/262, 95% CI: 4.2%−11.1%) of the included pregnant women reported use of tobacco products. The tobacco products that were reportedly used in this study included tobacco cigarettes (7.3%, 19/262, 95% CI: 3.8%−10.7%) and tobacco heating systems (0.4%, 1/262, 95% CI: 0.0%−3.8%). Tobacco cigarettes were used along with ENDS in 0.8% (2/262, 95% CI: 0.0–3.4%) and with NRT in 0.4% (1/262, 95% CI: 0.0–3.8%) of the pregnant women in the survey.

Cannabinoid use was reported among 3.8% (10/262, 95% CI: 1.1%−7.01%) of the pregnant women. All the cannabinoid product users reported using products containing CBD only, and no one used products with THC > 1%. No concomitant use of tobacco and cannabinoids was reported among the pregnant women. The results are presented in Table 1.

### Prevalence of use of tobacco, ENDS, NRT, and cannabinoids: stratified by trimester of pregnancy

When stratified by trimester, 10.3% (12/117,95% CI: 5.1%−16.1%) of pregnant women in their second trimester reported using tobacco products, while only 5.5% (8/145, 95% CI: 1.4%−9.9%) of the pregnant women in their third trimester reported any such use. Among pregnant women in their second trimester, all tobacco users were tobacco cigarette smokers, with 1.7% (2/117,95% CI:0.0%−7.9%) using tobacco cigarettes along with ENDS and 0.9% (1/117,95% CI:0.0%−7.1%) reporting use of tobacco cigarettes with NRT. In the case of pregnant women in their third trimester, 4.8% (7/145, 95% CI:0.7%−9.1%) of pregnant women reported smoking tobacco cigarettes and 0.7% (1/145, 95% CI:0.0%−4.9%) reported using tobacco heating system. No use of ENDS or NRT was reported among the included pregnant women in their third trimester. Cannabinoid use (CBD only) was reported by 5.1% (6/117,95% CI: 0.1%−9.6%) of pregnant women in their second trimester and by 2.8% (4/145,95% CI: 0.0%−6.9%) in their third trimester. The results are shown in Table 2.

### Frequency and prevalence of use of different types of tobacco products, ENDS, and NRT among pregnant users

Among pregnant women reporting tobacco use, 90.0% (18/20, 95% CI: 85%−100%) reported smoking conventional cigarettes, and 50% of them used less than 5 cigarettes per day. Roll-on cigarettes were used by 5.0% (1/20, 95% CI: 0.0%−19.8%) of the tobacco users and 5.0% (1/20, 95%CI: 0.0%−19.8%) also reported using tobacco heating systems. Among the ENDS and NRT users, 2 (50.0%,95% CI: 25.0%−100.0%) pregnant women reported using ENDS, and 1 (25%, 95% CI: 25.0%−86.0%) pregnant woman reported use of NRT. The results are summarized in Tables 3, 4.

### Intention to quit/reduce/not change the use of tobacco

On questioning the pregnant women who reported tobacco cigarette use about their intention to quit or reduce their current use, it was observed that 31.6% (6/19,95% CI: 10.5%−54.7%) wanted to quit, 36.8% (7/19,95% CI: 15.8%−60%)

**Table 1. Prevalence of tobacco, electronic nicotine delivery systems, nicotine replacement therapy, and cannabinoid use in pregnancy.**

| Prevalence of tobacco, electronic nicotine delivery systems (ENDS), nicotine replacement therapy (NRT), and cannabinoid use in pregnancy | N = 262 | Proportion | 95% confidence interval (CI) |
|---|---|---|---|
| Non-use of tobacco, ENDS, NRT, and cannabinoids | 221 | 84.4% | 80.5%−88.8% |
| Missing data on the use of tobacco, ENDS, NRT, and cannabinoids | 11 | 4.2% | 3.8%−8.6% |
| Tobacco use[a] | 20 | 7.6% | 4.2%−11.1% |
| Tobacco cigarettes (Conventional or roll-on cigarettes) | 19 | 7.3% | 3.8%−10.7% |
| with ENDS | 2 | 0.8% | 0.0%−3.4% |
| with NRT | 1 | 0.4% | 0.0%−3.8% |
| Tobacco heating systems | 1 | 0.4% | 0.0%−3.8% |
| with other products[b] | 1 | 0.4% | 0.0%−3.1% |
| Cannabidiol use (Exclusive cannabidiol (CBD) users)[c] | 10 | 3.8% | 1.1%−7.0% |

**Footnotes**

[a]Information on tobacco use was not available for 9 participants and for 1 participant the interview was not possible.

[b]Other products that were not specified in the survey.

[c]Information on cannabinoid use was not available for 10 participants and for 1 participant the conversation on cannabinoid use was not possible during the survey.

**Table 2. Prevalence of tobacco, electronic nicotine delivery systems, nicotine replacement therapy, and cannabinoid use in pregnancy stratified based on trimester of pregnancy.**

| Prevalence of tobacco, electronic nicotine delivery systems (ENDS), nicotine replacement therapy (NRT), and cannabinoid use in pregnancy | Second Trimester (N = 117) | | | Third Trimester (N = 145) | | |
|---|---|---|---|---|---|---|
| | N | Proportion | 95% confidence interval (CI) | N | Proportion | 95% confidence interval (CI) |
| Non-use of tobacco, ENDS, NRT, and cannabinoids | 95 | 81.2% | 75.2%−88.5% | 126 | 86.9% | 82.1%−92.0% |
| Missing data on the use of tobacco, ENDS, NRT, and cannabinoids | 4 | 3.4% | 0.0%−22.7% | 7 | 4.8% | 0.0%−9.9% |
| Tobacco use[a] | 12 | 10.3% | 5.1%−16.1% | 8 | 5.5% | 1.4%−9.9% |
| Tobacco cigarettes (Conventional or roll-on cigarettes) | 12 | 10.3% | 5.1%−16.1% | 7 | 4.8% | 0.7%−9.1% |
| with ENDS | 2 | 1.7% | 0.0%−7.9% | 0 | – | – |
| with NRT | 1 | 0.9% | 0.0%−7.1% | 0 | – | – |
| Tobacco heating systems | 0 | – | – | 1 | 0.7% | 0.0%−4.9% |
| with other products[b] | 0 | – | – | 1 | 0.7% | 0.0%−4.9% |
| Cannabidiol use (Exclusive cannabidiol (CBD) users)[c] | 6 | 5.1% | 0.1%−9.6% | 4 | 2.8% | 0.0%−6.9% |

**Footnotes**

[a]Among the women in their second trimester, three had missing data on tobacco use due to unavailable information. In the third trimester, six women had missing data on tobacco use; for five of them, the information was not available, and for one, the interview could not be conducted.

[b]Other products that were not specified in the survey.

[c]Regarding cannabinoid use, among the participants in their second trimester, data were unavailable for three women, while one could not be interviewed during the survey. For those in their third trimester, seven participants had missing data on cannabinoid use, with no information available for any of them

**Table 3. Prevalence and Frequency of use of tobacco in pregnancy.**

| Tobacco products | Tobacco Users (N = 20) | | Frequency (Number/day) N (%) | | | | | |
|---|---|---|---|---|---|---|---|---|
| | N (%) | 95%Confidence interval (CI) | <5 | 6-10 | 11-15 | >15 | Other | No information |
| Conventional Cigarette | 18(90.0%) | 85.0%−100.0% | 9(50.0%) | 5 (27.8%) | 0 | 1 (5.6%) | 0 | 3 (16.7%) |
| Roll-on Cigarette | 1(5.0%) | 0.0%−19.8% | 0 | 1 (100.0%) | 0 | 0 | 0 | 0 |
| Tobacco heating system | 1(5.0%) | 0.0%−19.8% | 0 | 0 | 0 | 0 | 0 | 1(100.0%) |

**Table 4. Prevalence and Frequency of use of electronic nicotine delivery systems, nicotine replacement therapy, and other products in pregnancy.**

| Electronic nicotine delivery systems (ENDS), nicotine replacement therapy (NRT), and other products use in pregnancy | Number of participants (N = 4) | | Frequency (Number/day)/ (ml/day [for liquids]) N (%) | | | | | |
|---|---|---|---|---|---|---|---|---|
| | N (%) | 95%Confidence interval (CI) | <5 | 6-10 | 11-15 | >15 | Other | No information |
| ENDS users | | | | | | | | |
| ENDS without nicotine | 2(50.0%) | 25.0%−100.0% | 0 | 2 (100.0%) | 0 | 0 | 0 | 0 |
| NRT users | | | | | | | | |
| Inhaler | 1(25.0%) | 0.0%−86.0% | 0 | 1 (100.0%) | 0 | 0 | 0 | 0 |
| Other product users | 1(25.0%) | 0.0%−86.0% | 0 | 0 | 0 | 0 | 0 | 1 (100.0%) |

wanted to reduce, and 26.3% (5/19,95% CI: 5.3%−49.5%) wanted no change to the current use of tobacco cigarettes. However, one of two ENDS users reported an intention to quit smoking, while the only NRT user also reported her intention to reduce tobacco smoking. The results are presented in Table 5.

### Prevalence of smoking cessation counseling intervention in pregnancy

Only 25% (5/20, 95% CI: 10.0%−48.3%) of those who used tobacco products in pregnancy had received a smoking cessation counseling intervention, and all those who received this intervention reported smoking tobacco cigarettes. Only one among those who received the smoking cessation intervention was recommended to use NRT as an alternative. None received a recommendation for pharmacotherapy. The results are presented in Table 6.

## Discussion

In this cross-sectional survey of 262 pregnant women coming for their regular clinical visits in one center in Switzerland, we found a prevalence of tobacco products at 7.6%, ENDS at 0.8%, NRT at 0.4%, and cannabinoid use at 3.8%.

Previously conducted studies observed the prevalence of tobacco use among pregnant women in Switzerland to vary between 7.0% and 21.0% [47–49]. This is in line with the current study showing a prevalence of 7.6%. The prevalence of tobacco use observed in our study is, however, lower than that observed in the National perinatal survey conducted in France. In the French perinatal survey, 16.6% of women smoked tobacco in their third trimester of pregnancy in 2016, which then decreased to 12.2% in 2021 [50–52]. However, the observed difference could stem from the social and cultural differences around smoking in both the countries, particularly with the strong cultural legacy of smoking in France [53]. Nevertheless, more studies are necessary to understand how various social, cultural, and economic factors influence the observed disparity in smoking prevalence between France and Switzerland to the fullest extent. Additionally, upon stratifying tobacco use by trimester, we observed that the number of tobacco product users in the second trimester was nearly twice as high as in the third trimester. It is, however, not clear if this difference is due to ceasing tobacco use fearing adverse effects during later stages of pregnancy, due to underreporting, or chance [54,55]. While assessing the

**Table 5. Preference to quit/change tobacco use among tobacco, electronic nicotine delivery systems, nicotine replacement therapy, other products users in pregnancy.**

| Preference | Conventional or roll-on cigarette users (N=19) | | Tobacco heating system users (N=1) | | Electronic nicotine delivery systems (ENDS) users (N=2) | | Nicotine replacement therapy (NRT) user(N=1) | | Other product user[d] (N=1) | |
|---|---|---|---|---|---|---|---|---|---|---|
| | N (%) | 95% CI | N (%) | 95% CI | N (%) | 95% CI* | N (%) | 95% CI | N (%) | 95% CI |
| Want to quit | 6(31.6%) | 10.5%−54.7% | 0 | – | 1(50.0%) | 50.0%−100.0% | 0 | – | 0 | – |
| Wants to reduce | 7(36.8%) | 15.8%−60% | 0 | – | – | – | 1(100%) | – | 0 | – |
| No change | 5(26.3%) | 5.3%−49.5% | 1 (100.0%) | – | 1 (50.0%) | 50.0%−100.0% | 0 | – | 1 (100%) | – |
| Missing[a] | 1(5.3%) | 0.0%−28.4% | 0 | – | – | – | 0 | – | 0 | – |
| Interested in a study to switch to ENDS[b] | 1(5.3%) | 0.0%−20.8% | 0 | – | 0 | – | 0 | – | 0 | – |
| Missing[c] | 1(5.3%) | 0.0%−20.8% | 0 | – | 0 | – | 0 | – | 0 | – |

**Footnotes**

[a]No information was available for these participants.

[b]This variable was only recorded for those who used tobacco [conventional/roll-on cigarette or tobacco heating system (IQOS)].

[c]Only 1 participant had missing information on interest in the study to switch to e-cigarettes.

[d]Other products that were not specified in the survey.

CI: Confidence Interval

**Table 6. Smoking cessation intervention for pregnant tobacco product users and alternatives suggested.**

| Smoking cessation intervention | Tobacco users (N = 20) | |
|---|---|---|
| | N (%) | 95% Confidence Interval (CI) |
| Received a smoking cessation counseling[a] | 5 (25.0%) | 10.0% −48.3% |
| Suggested Alternative | | |
| Nicotine replacement therapy (NRT) | 1(5.0%) | 0.0-27.8% |
| Not specified | 1(5.0%) | 0.0-27.8% |
| Missing[b] | 3 (15.0%) | 0.0%−37.8% |
| Missing[c] | 2 (10.0%) | 0.0%−33.3% |

**Footnotes**

[a]All tobacco users who received smoking cessation counseling were conventional or roll-on cigarette users.

[b]No information was available for these individuals regarding the alternative suggested.

[c]No information was available for these individuals on whether they had received smoking cessation counseling.

prevalence of cannabinoid use, none of the pregnant women reported using cannabinoid products with THC > 1% but reported use of only CBD products. This could be due to potential underreporting of THC use, as the use of cannabinoid products with THC > 1% for recreational purposes is to date prohibited in Switzerland [56]. Additionally, unlike THC, CBD is less psychoactive and thus perceived to be "safe" when compared to THC [35]. Nevertheless, there is a paucity of evidence on the safety of using CBD in pregnancy. Preclinical studies have pointed out potential gender-specific deficits in problem-solving, motor, and discriminatory abilities [57,58]. However, more studies with real-world data are necessary to fully understand the effect of CBD on perinatal outcomes. Like tobacco use, when stratified by trimester, the number of CBD product users in the third trimester was approximately half of that observed in the second trimester. In Switzerland, the use of CBD products containing 3–20% CBD and THC < 1% for smoking is increasing [59,60]. A survey conducted among the general population in Switzerland has highlighted the co-use of tobacco, in addition to e-cigarettes and cannabis among CBD users [61]. Although no co-use of these products was observed in our study, this could still be due to potential underreporting. Additionally, the form or route of consumption of CBD was not collected in this study.

On assessing their intention to quit smoking, 32% of pregnant women in this survey wanted to quit smoking tobacco cigarettes. Observational studies conducted in the past found a significant association between quitting tobacco use during pregnancy and a positive impact on birth weight & neonatal body composition in infants [14,62–65]. A cross-sectional study on live births between 2011 and 2017 observed that women who quit smoking after knowing about pregnancy or during the first trimester had a lower risk of preterm birth compared to those who smoked throughout the pregnancy [66]. Nevertheless, 37% of tobacco cigarette smokers in this survey sample wanted to reduce the use of tobacco in pregnancy. Although quitting tobacco use during pregnancy provides numerous benefits and should be the primary goal, reducing tobacco use can still have positive effects for women who find it difficult to quit. Therefore, they should be encouraged to reduce their tobacco use [14]. Around 26% of pregnant women smoking cigarettes in this survey sample wanted no change in the use of tobacco. Educating this subgroup of pregnant women on the merits of smoking cessation by healthcare providers could be beneficial [67].

In our study, a small proportion of women reported ENDS (0.8%) and NRT (0.4%) use. Among them, one ENDS user mentioned her intent to quit using tobacco, and the sole NRT user also intended to reduce smoking. Additionally, NRT was also suggested as an alternative to support smoking cessation for one pregnant woman who reportedly smoked tobacco and also received smoking cessation counseling in our study. In addition, none were suggested with pharmacotherapy as

an alternative. ENDS, NRT, and pharmacotherapy with Varenicline or Bupropion are reportedly used for smoking cessation in pregnancy and can help mitigate the harmful effects associated with tobacco [25,68]. An RCT assessing the impact of the use of NRT in pregnancy observed a significant reduction in the number of smoked cigarettes among pregnant women and an increase in birth weight and gestational age with NRT use [69]. The Pregnancy trial of E-cigarettes and Patches (PREP) study, a recently conducted RCT, highlighted that the use of ENDS and NRT can improve smoking cessation without increasing the risk of adverse outcomes in pregnancy when compared to abstainers who do not use such products [70]. This study also observed no difference in the risk of adverse neonatal outcomes such as low birth weight, and NICU (neonatal intensive care unit) admission between abstainers using ENDS or NRT during pregnancy and abstainers using no such aids [70]. Additionally, abstainers using ENDS or NRT had 78% reduced risk of preterm birth compared to abstainers using no aids, during pregnancy [70]. Although ENDS and NRT are seen to have similar safety profiles in pregnancy, an RCT conducted in England observed, that ENDS does not increase the risk of adverse birth outcomes when compared to NRT and also reduced the risk of low birth weight by 35% when compared to those using NRT in pregnancy [18]. A recently published retrospective cohort study also observed that the use of NRT, Varenicline, or Bupropion during pregnancy was not associated with an increased risk of major congenital malformations compared to smoking during the first trimester. Although more research is required to assess the effectiveness and safety of NRT and pharmacotherapy use in pregnancy, providing cessation support using these resources could be beneficial in reducing the adverse outcomes due to smoking in pregnancy [14,71,72].

It is noteworthy that, despite many pregnant tobacco users willing to quit or reduce tobacco use, only 25% of pregnant tobacco users received smoking cessation counseling in our study. This could be due to the potential underreporting of tobacco use in pregnancy to healthcare providers. Underreporting of tobacco use in pregnancy could arise due to the stigma and stereotypes associated with it [54]. The stigma associated with smoking could motivate women to conceal smoking and sometimes encourage them to binge smoke, which could have harmful effects on the mother and fetus [73]. The fact that a considerable proportion of pregnant women smoking cigarettes reported their intentions to quit (32%) or reduce (37%) tobacco use in this anonymous survey highlights the potential need for an intervention that effectively maintains patient anonymity to support smoking cessation efforts. A digital intervention, such as a smartphone app for smoking cessation, could be an effective solution, as it is cost-effective, highly flexible, and preserves patient anonymity [74]. Further studies should be conducted to fully understand the potential for a personalized digital intervention for smoking cessation in pregnant women. In our anonymized survey, 4.2% of included pregnant women still did not provide any response when questioned on their tobacco, ENDS, NRT, and cannabinoid use. However, these participants had provided data on age and gestational week and were therefore retained in the analysis to reflect the presence of individuals who may be reluctant to disclose sensitive behaviors. To assess the potential impact of non-response on our estimates, we conducted a sensitivity analysis assuming that all non-respondents reported tobacco use during pregnancy. Under this assumption, the prevalence of tobacco use among pregnant women in our sample would increase to 11.8%. This suggests a potential underestimation in our primary prevalence estimates due to non-response. Nevertheless, this estimate from sensitivity analysis is still within the prevalence range of 7.0%−21.0%, reported from previous studies estimating the prevalence of smoking among pregnant women in Switzerland, and lower than that observed in the French perinatal survey. Using digital intervention to collect information on tobacco and cannabinoid use, while preserving the anonymity of the patient might improve the response rate.

## Strengths and limitations

In contrast to previous studies, this study assessed the frequency of use of tobacco, ENDS, NRT, and cannabinoid products stratified by pregnancy trimester. Notably, it is among the few studies that also examined cannabinoid use among pregnant women. Nonetheless, the study has several limitations. This is a single-centered study with a small sample size, and key information on predictors of substance use and cessation such as education, employment status, relationship

status, the socioeconomic and psychological characteristics of the women included in the study was not collected. Thus, it was not possible to assess if our study cohort was a representative sample of the pregnant population in Switzerland or a high-risk group. Due to the cross-sectional design of the study, it was not possible to collect information on if tobacco users tend to quit tobacco or switch to using ENDS/NRT over time in pregnancy. Another important limitation is that tobacco and cannabinoid use in this survey has been self-reported, which could also be prone to underreporting, thus resulting in an underestimation of the prevalence of such use in our sample. Future studies could incorporate additional laboratory tests to validate the anonymized self-report, based on the cost and feasibility of such tests, to accurately identify the users. In the case of cannabinoid use, legal constraints on THC use in Switzerland could also have resulted in the underreporting of THC use in pregnancy, resulting in an underestimation of its prevalence. Also, the mode of consumption (vaping/topical/oral) of CBD was not collected, which limits the knowledge on the potential co-use of tobacco by pregnant women. Thus, the result of this study needs to be interpreted with caution.

## Conclusion

This monocentric cross-sectional survey conducted in Switzerland focused on assessing the proportion of tobacco, ENDS, NRT, and cannabinoid use among pregnant women at a hospital for regular check-ups in Biel, Switzerland. The results showed that 7.6% of pregnant women used tobacco products. ENDS and NRT were reportedly used by 0.8% and 0.4% of the pregnant women, and 3.8% of the pregnant women reported the use of cannabinoids, particularly products containing only CBD during pregnancy. Although many pregnant tobacco users were willing to reduce or quit smoking, only 25% of them received smoking cessation counseling. The use of digital intervention could improve reporting of tobacco use and promote participation in smoking cessation counseling. Future studies should, however, evaluate the potential of such digital strategies. Additionally, further research is required to collect comprehensive information on the form of cannabinoid use among pregnant women in Switzerland to understand the co-use of tobacco among cannabinoid users.

## Acknowledgments

We thank all the pregnant women for their participation, all the midwives for their help in collecting the data, and the clinic for their willingness to participate in the survey.

## Author contributions

**Conceptualization:** Anna Schoeni.

**Formal analysis:** Sreemanjari Kandhasamy, Stéphanie Baggio, Alice Panchaud, Stephen P. Jenkinson, Anna Schoeni.

**Funding acquisition:** Reto Auer.

**Investigation:** Jérôme Mathis, Yolanda Mattmann.

**Methodology:** Sreemanjari Kandhasamy, Stéphanie Baggio, Emeline Maisonneuve, Reto Auer, Alice Panchaud, Stephen P. Jenkinson, Anna Schoeni.

**Supervision:** Stéphanie Baggio, Alice Panchaud, Stephen P. Jenkinson, Anna Schoeni.

**Visualization:** Sreemanjari Kandhasamy.

**Writing – original draft:** Sreemanjari Kandhasamy, Stéphanie Baggio, Alice Panchaud, Stephen P Jenkinson, Anna Schoeni.

**Writing – review & editing:** Sreemanjari Kandhasamy, Stéphanie Baggio, Jérôme Mathis, Yolanda Mattmann, Emeline Maisonneuve, Reto Auer, Alice Panchaud, Stephen P. Jenkinson, Anna Schoeni.

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
