## [Decision Letter · Decision Letter 0]

3 Jun 2025

Dear Dr. Schoeni,

Thank you for submitting your manuscript to PLOS ONE. After careful consideration, we feel that it has merit but does not fully meet PLOS ONE’s publication criteria as it currently stands. Therefore, we invite you to submit a revised version of the manuscript that addresses the points raised during the review process.

**ACADEMIC EDITOR:**
**Please address undermentioned reviewer points and resubmit for reconsideration. **

We look forward to receiving your revised manuscript.

Kind regards,

Souparno Mitra, M.D.

Academic Editor

PLOS ONE

Journal Requirements:

For additional information about PLOS ONE ethical requirements for human subjects research, please refer to http://journals.plos.org/plosone/s/submission-guidelines#loc-human-subjects-research .

6. In the online submission form, you indicated that the data used and analyzed in this study may be made available upon reasonable request to the corresponding author.

7. We note that you have indicated that there are restrictions to data sharing for this study. For studies involving human research participant data or other sensitive data, we encourage authors to share de-identified or anonymized data. However, when data cannot be publicly shared for ethical reasons, we allow authors to make their data sets available upon request. For information on unacceptable data access restrictions, please see http://journals.plos.org/plosone/s/data-availability#loc-unacceptable-data-access-restrictions.

8. Please update your submission to use the PLOS LaTeX template. The template and more information on our requirements for LaTeX submissions can be found at http://journals.plos.org/plosone/s/latex.

9 . Please review your reference list to ensure that it is complete and correct. If you have cited papers that have been retracted, please include the rationale for doing so in the manuscript text, or remove these references and replace them with relevant current references. Any changes to the reference list should be mentioned in the rebuttal letter that accompanies your revised manuscript. If you need to cite a retracted article, indicate the article’s retracted status in the References list and also include a citation and full reference for the retraction notice.

Reviewers' comments:

Reviewer's Responses to Questions

**Comments to the Author**

1. Is the manuscript technically sound, and do the data support the conclusions?

Reviewer #1: Yes

Reviewer #2: Yes

2. Has the statistical analysis been performed appropriately and rigorously?

Reviewer #1: Yes

Reviewer #2: I Don't Know

3. Have the authors made all data underlying the findings in their manuscript fully available?

Reviewer #1: Yes

Reviewer #2: No

4. Is the manuscript presented in an intelligible fashion and written in standard English?

Reviewer #1: Yes

Reviewer #2: Yes

Reviewer #1: Reviewed the article- "Tobacco, electronic nicotine delivery system, nicotine replacement therapy, and cannabinoid use during pregnancy: A descriptive cross-sectional survey." Attached are the suggested edits.

Reviewer #2: The study "Tobacco, electronic nicotine delivery system, nicotine replacement therapy, and cannabinoid use during pregnancy: A descriptive cross-sectional survey" addresses substance use during pregnancy—a critical concern given the risks to maternal and fetal health—and expands to include CBD, ENDS, and NRT. The survey captures the self-reported intention to quit or reduce tobacco use, which is useful for intervention development.

Please add these limitations to the study:

1. Key predictors of substance use and cessation (education, employment, relationship status, mental health) were not collected, limiting context and interpretation.

2. A snapshot design limits understanding of whether women quit later in pregnancy or switched substances—insights useful for tailoring interventions.

3. While missing responses are reported, the potential bias from non-response (4.2% on substance use) is not analyzed or statistically accounted for. Please explain.

4. The mode of use (e.g., vaping vs. oral vs. topical) affects fetal exposure risk, but this was not assessed.

**Do you want your identity to be public for this peer review?** For information about this choice, including consent withdrawal, please see our Privacy Policy

Reviewer #1: **Yes: ** Arun Prasad

Reviewer #2: **Yes: ** Nikhil Tondehal

---

## [Author Response · Author response to Decision Letter 1]

8 Aug 2025

We thank the reviewers and the editor for their inputs on our manuscript. In response to the reviewer's comments, we have now elaborated and refined our limitation section, conducted a minor sensitivity analysis for the missing responses, and also provided potential explanations for the observed difference in prevalence of smoking in pregnancy between Switzerland and France. Kindly find the detailed rebuttal letter responding to all the comments of the reviewers and editor attached.

We have also updated ethics, data availability, and the financial disclosure statements ( presented in cover letter and rebuttal letter). The same has been also presented in the rebuttal letter. I have now updated the ethics and data availability statement online, however I could not find a place to enter the updated financial disclosure statement online, but this however is presented in the cover letter and in the rebuttal letter.

---

## [Decision Letter · Decision Letter 1]

7 Sep 2025

Tobacco, electronic nicotine delivery system, nicotine replacement therapy, and cannabinoid use during pregnancy: A descriptive cross-sectional survey

PONE-D-25-19369R1

Dear Dr. Schoeni,

We’re pleased to inform you that your manuscript has been judged scientifically suitable for publication and will be formally accepted for publication once it meets all outstanding technical requirements.

Kind regards,

Souparno Mitra, M.D.

Academic Editor

PLOS ONE

Additional Editor Comments (optional):

Reviewer #1:

Reviewer #2:

Reviewers' comments:

Reviewer's Responses to Questions

**Comments to the Author**

Reviewer #1: All comments have been addressed

Reviewer #2: All comments have been addressed

2. Is the manuscript technically sound, and do the data support the conclusions?

Reviewer #1: Partly

Reviewer #2: Yes

3. Has the statistical analysis been performed appropriately and rigorously?

Reviewer #1: I Don't Know

Reviewer #2: Yes

4. Have the authors made all data underlying the findings in their manuscript fully available?

Reviewer #1: Yes

Reviewer #2: Yes

5. Is the manuscript presented in an intelligible fashion and written in standard English?

Reviewer #1: Yes

Reviewer #2: Yes

Reviewer #1: Thank you for making suggested edits to the article "Tobacco, electronic nicotine delivery system, nicotine replacement therapy, and cannabinoid use during pregnancy: A descriptive cross-sectional survey"

Reviewer #2: (No Response)

**Do you want your identity to be public for this peer review?** For information about this choice, including consent withdrawal, please see our Privacy Policy

Reviewer #1: **Yes: ** Arun Prasad

Reviewer #2: **Yes: ** Nikhil Tondehal

---

## [Editor Report · Acceptance letter]

PONE-D-25-19369R1

PLOS ONE

Dear Dr. Schoeni,

I'm pleased to inform you that your manuscript has been deemed suitable for publication in PLOS ONE. Congratulations! Your manuscript is now being handed over to our production team.

Kind regards,

on behalf of

Dr. Souparno Mitra

Academic Editor

PLOS ONE